# A Novel Wearable EEG and ECG Recording System for Stress Assessment

**DOI:** 10.3390/s19091991

**Published:** 2019-04-28

**Authors:** Joong Woo Ahn, Yunseo Ku, Hee Chan Kim

**Affiliations:** 1Seoul National University Hospital Biomedical Research Institute, Seoul 03082, Korea; kooljungwoo@snu.ac.kr; 2Department of Biomedical Engineering, College of Medicine, Chungnam National University, Daejeon 35015, Korea; 3Interdisciplinary Program in Bioengineering, Graduate School, Seoul National University, Seoul 03080, Korea; hckim@snu.ac.kr; 4Department of Biomedical Engineering, College of Medicine, Seoul National University, Seoul 03080, Korea; 5Institute of Medical & Biological Engineering, Medical Research Center, Seoul National University, Seoul 03080, Korea

**Keywords:** stress assessment, wearable device, heart rate variability, electroencephalogram

## Abstract

Suffering from continuous stress can lead to serious psychological and even physical disorders. Objective stress assessment methods using noninvasive physiological responses such as heart rate variability (HRV) and electroencephalograms (EEG) have therefore been proposed for effective stress management. In this study, a novel wearable device that can measure electrocardiograms (ECG) and EEG simultaneously was designed to enable continuous stress monitoring in daily life. The developed system is easily worn by hanging from both ears, is lightweight (i.e., 42.5 g), and exhibits an excellent noise performance of 0.12 μVrms. Significant time and frequency features of HRV and EEG were found in two different stressors, namely the Stroop color word and mental arithmetic tests, using 14 young subjects. Stressor situations were classified using various HRV and EEG feature selections and a support vector machine technique. The five-fold cross-validation results obtained when using both EEG and HRV features showed the best performance with an accuracy of 87.5%, which demonstrated the requirement for simultaneous HRV and EEG measurements.

## 1. Introduction

Stress has become a serious problem in modern society and can lead to illnesses such as cognitive dysfunction, depression, and even cardiovascular disease [1,2]. If stress becomes chronic, it weakens the human immune system allowing easy infection and delayed recovery processes [3]. Stress affects not only our physical health but also our work performance, passion for work, and general attitude in daily life [4]. It has been reported that an increase in employee stress levels in the workplace reduces a company’s overall performance, which can lead to economic burdens on society [5,6,7]. As such, prevention-oriented stress management is necessary for individual health, as well as for the welfare of society.

Stress is known to induce abnormal responses in the autonomic nervous system (ANS), which consists of the sympathetic nervous system (SNS) and the parasympathetic nervous system (PNS) under antagonistic control [8,9]. These two systems are related to stress and relaxation reactions, respectively, so that stress activates the SNS and suppresses the PNS [10,11]. In this context, the heart rate variability (HRV), i.e., the variation in the time interval between heartbeats, is known to be a reliable noninvasive biomarker of the ANS [12]. The HRV typically increases during relaxing activities and decreases during stress. Indeed, many previous studies have used the HRV to assess ANS activities in response to mental stress [13,14,15,16,17,18,19]. In addition, electroencephalograms (EEG), which reflect brain activity, are also important for detecting and assessing mental stress [20,21,22]. Some neurophysiological studies have reported the relationship between human emotion and hemispheric specialization [23,24], where the left hemisphere is more involved in processing positive emotions, and the right hemisphere is more involved in processing negative emotions. Furthermore, the prefrontal cortex accounts for a large proportion of emotional processing [25]. Stress usually causes negative moods, such as depression, anger, and anxiety, resulting in increased right-prefrontal activity [26]. Thus, the asymmetric analysis of the frequency-band powers in the EEG measured at the prefrontal cortex has been generally applied in previous stress studies [20,26,27,28,29,30]. Moreover, the simultaneous analysis of both the HRV and EEG is likely to provide a more precise assessment of stress [31] compared to the analysis of HRV alone, and one of the main objectives of this study is to verify this hypothesis.

In terms of achieving continuous stress management in daily life, device usability is key. For EEG measurements, the majority of clinical studies have used EEG channels from hair-bearing scalp areas of the 10–20 system. However, this method requires the use of a conductive gel and an appropriate preparation procedure, which are particularly inconvenient for users [32]. Indeed, EEG recordings from hairless regions such as the forehead, or behind or inside the ear, would be more suitable for long-term monitoring in daily life [32,33]. In terms of the HRV measurements, electrocardiograms (ECG) from two or more electrodes on the chest or wrist are commonly employed. However, to the best of our knowledge, it is currently necessary to use two different devices for simultaneous EEG and ECG recordings.

Thus, we herein report the development of a wearable device that can measure left and right EEG signals and a channel ECG signal simultaneously with only three electrodes behind the ears and one electrode on the forehead. The analog front-end (AFE) is designed to share active electrodes for the EEG and ECG signals, and simultaneous recording of both signals will be verified through the eye open/closed EEG test and the conventional lead I ECG reference. Furthermore, stress experiments will be performed to test the feasibility of the proposed system in stress assessment. The Stroop color word test [34,35] and a mental arithmetic test [36], which have been widely used in many studies, will be applied as the stressors. Statistical analysis will also be performed on the HRV and EEG parameters to extract significant features for the classification of stressor situations. Finally, the results of the classification performances for different feature selections will be evaluated.

## 2. Materials and Methods

### 2.1. System Design

Since the amplitude of the ECG and EEG measured in the head are particularly small (i.e., tens of μV), the wearable system was designed using a low-noise circuit. We also focused on a low power consumption and a small-sized circuit, such that long-term and everyday use would be possible.

We included an analog front-end (AFE) for amplification and filtering of the raw EEG and ECG data to improve the signal-to-noise ratio. The AFE is comprised of an instrument amplifier for differential amplification, a high-pass filter (HPF), and a low-pass filter (LPF). For the HPF and LPF, a second-order Butterworth filter (roll-off = 12 dB/octave), which uses the Sallen–Key topology, was designed. The cut-off frequencies of the HPF and LPF were set to 0.5 Hz and 50 Hz, respectively. An ADS1294 instrument (Texas Instruments Inc., Dallas, TX, USA) was used as the analog to digital converter (ADC). This instrument is composed of four ADC channels and is an IC manufactured to measure the biopotential. A PBLN51822 Bluetooth low-energy module (Prochild Inc., Seoul, Korea) was employed to give a low-power system, and an ATmega168 microcontroller (Microchips Technology Inc., Chandler, AZ, USA) was used for the system control. The overall circuit diagram of the EEG and ECG recording system is shown in Figure 1. 

When selecting the electrode positioning, the EEG and HRV must be measured simultaneously to accurately assess stress, and the electrodes must be included in the wearable system to improve the system’s ease of wear. The EEG features used to assess stress were the band powers and band power asymmetries of the left and right hemispheres, and so the active electrodes were placed at the bottom of the mastoids on both sides, such that the EEGs of the left and right hemispheres could be measured, as shown in Figure 2. The reference electrode was placed on the forehead, and the ground electrode was placed on the left mastoid. As shown in Figure 2, if the active 1 electrode and the active 2 electrode are connected to the differential input of the INA, the ECG can be obtained. Therefore, not only can the HRV be extracted from the measured ECG, but no additional electrodes are required for the ECG measurements.

### 2.2. EEG and ECG Recording Experiment

Seven healthy young male subjects (mean age: 29.3 ± 2.4) participated in the experiments after signing the informed consent forms. The subjects performed the tests by sitting on a comfortable chair in a typical office. Skin abrasion was not performed before attaching the electrodes. Experiments were performed to verify that the EEG and ECG were measured effectively by the developed wearable system. In addition, the HRV extracted from the ECG signal measured at the head was verified. In the case of the EEG, a significantly large alpha can be measured and verified when most people close their eyes, and this has been used in a number of studies. The alpha wave was measured in both the left and right sides of the hemispheres, and this is an especially important indicator for the assessment of stress. The subjects participating in the experiments performed the tests for a total of 80 s and repeatedly opened and closed their eyes every 20 s. ECGs were measured at the head and both arms simultaneously, which is a standard measurement method, and the ECGs were compared. The ECG measured on both arms was a lead I, which showed the potential of the heart’s left and right sides; since a complete QRS waveform could be observed, it was considered sufficient for extracting the HRVs [37], which were used as stress features. Thus, the HRVs were extracted using the ECG measured from the standard position and the head, and their accuracies were compared. The subjects participating in the experiments performed the tests until 30 R-peaks were measured.

### 2.3. Stress Experiments

Fourteen healthy young male subjects (mean age: 29.4 ± 3.3) participated in the stress test. The stress test was reviewed and approved by the IRB of the Seoul National University Hospital Biomedical Research Institute (IRB. No. H-1709-010-881). The recruited subjects had not experienced any disease or surgery that could affect their HRVs. Furthermore, subjects who were sleepy, excessively stressed, or had consumed alcohol or medication before the tests were excluded. The Stroop color word test and a mental arithmetic test were used as methods for inducing stress. The Stroop color word test is a reliable and valid method for inducing medium levels of stress [34,35]. In this test, the names of colors are shown in colors that are different from the colors corresponding to their names, and the subject must pick the colors that the names appear in (see Figure 3a). A total of 56 test stimuli was used, and they were each displayed randomly for 1 s. Because speaking can affect the HRV, the subjects were asked to think about the correct color, and the answer was displayed on the screen at the end of each stimulus. The mental arithmetic test is a standard moderate intensity stressor used in physiology to detect changes in the ANS function [36]. The subjects are shown numbers with three or four digits on the screen, and the digits are added together repeatedly until arriving at a one-digit number. Finally, the subject uses a keyboard to enter whether the final result is an even number or an odd number (see Figure 3b). To generate a greater amount of stress in the subjects, 5 s were counted for each stimulus prior to moving on to the next stimulus, and the screen displays whether the subject’s input was right or wrong for each stimulus.

All participants performed the tests in an office of regular temperature (23–25 °C), 350-lux brightness, and minimized visual or auditory stimuli that could affect stress levels. In addition, to prevent drowsiness, no tests were performed within 1 h of eating. The overall test protocol is outlined in Figure 4. Initially, the equipment was attached to the participant, and the participant sat in a comfortable chair and waited for 10 min. This was a stabilizing period to ensure that the subject’s EEG and ECG were output correctly, and also to stabilize any stress that may occur from attachment of the equipment or the posture of sitting on the chair. Subsequently, the EEG and ECG recordings began, and the tests were performed in the following order: Stroop test (S1)—rest (R1)—mental arithmetic test (S2)—rest (R2). Each test stage was performed over 6 min for the short-term HRV analysis, as recommended by HRV measurement standards [38]. This enabled assessment of the change in the sympathovagal balance, which appears directly after the beginning and end of the stress stimuli. The stress protocol was modified by referencing existing studies on stress assessment [13]. Figure 5 shows a participant taking part in the tests.

### 2.4. Data Analysis

Features were extracted for the stress assessment using the measured EEG of the left and right hemispheres in addition to the ECG. Signal processing for the extraction of each feature was performed using MATLAB 2017A (Mathworks Inc., Natick, MA, USA). In the obtained raw ECG data, the signal was divided according to each section (S1, R1, S2, R2), and the Pan and Tomkins algorithm [39] was used to obtain the R-peaks and the HRV. The HRV was lineally interpolated and resampled by 2 Hz to obtain uniformly spaced intervals, following which a fast Fourier transform (FFT) was performed to calculate the power spectrum for extracting the features in the frequency domain.

The calculated power spectrum was separated into low frequency (0.04–0.15 Hz) and high frequency (0.15–0.4 Hz) domains; the power of each was calculated, and the frequency features were extracted. After separating the obtained raw EEG data by section (S1, R1, S2, R2), band-pass filtering (1–35 Hz) was performed. The electrooculogram is one of the largest artifacts when the EEG signal is measured, and so it was removed prior to signal processing. EEGs that exceeded the 100 μV threshold were assumed to be EOG and were removed. FFT was used on the EEG with the removed EOG to calculate the power spectrum, and the features were extracted after obtaining the power of each band (delta, theta, alpha, beta). The features calculated in all frequency domains were normalized by dividing by the power of all frequency bands for each patient to preserve the variation between patients.

Table 1 shows the name, description, and equation of each feature extracted from the EEG and HRV for the stress assessment. The features extracted from the HRV include the mean of the R–R interval (mRR), the standard deviation of the R–R interval (SDRR), the root-mean-square difference of successive R–R intervals (RMSSD), the normalized low-frequency power of HRV (nLF-HRV), the normalized high-frequency power of HRV (nHF-HRV), and the ratio between nLF-HRV and nHF-HRV (LF/HF). The features extracted from the EEG include the left and right normalized alpha band power (nLAP, nRAP), the left and right normalized beta band power (nLBP, nRBP), and each band’s (delta, theta, alpha, beta) power asymmetry (DPA, TPA, APA, BPA).

To confirm the statistical significance of the features extracted from each section (S1, R1, S2, R2), an analysis of variance (ANOVA) was performed. ANOVA is a statistical technique that compares the differences between the means of groups, and it is useful when comparing three or more groups. ANOVA was performed via SPSS 21.0 (IBM Inc., Armonk, NY, USA), and the stressor situations were classified using the significant features and a support vector machine (SVM). An SVM is a machine-learning algorithm that is primarily used in classification and regression analysis. Since the main purpose of this study was to confirm the performance improvement of the stressor classification model when using both EEG and HRV features, additional machine learning algorithms were not applied. We also found that the SVM has been widely adopted in many previous stress assessment studies [7,40,41].

## 3. Results and Discussion

### 3.1. System Specifications

We designed the printed circuit board (PCB) shown in Figure 6. A PCB with rigid segments at both ends and in the middle was therefore constructed to optimize the space usage, and the parts were arranged above and below. The portions between the rigid PCB segments were fabricated using a flexible PCB such that it could be placed in a bendable wearable device (Figure 6a). In addition, since the current created in the digital part may interfere with the analog signal, the digital and analog lines, power planes, parts were divided [42]. The components present on the top and bottom of the rigid PCB are shown in Figure 6b,c, respectively. The PCB measured 350 × 18 × 1 mm^3^, the rigid PCB segments on the two ends measured 37 × 18 × 1 mm^3^, and the rigid PCB segment in the middle measured 30 × 16 × 1 mm^3^.

Figure 7a shows a photographic image of the developed wearable EEG and ECG system. Medical-use hydrogel electrodes (100 foam electrodes, Ludlow Technical Products Inc., Gananoque, ON, Canada) are attached to the body to measure the EEG and ECG. Figure 7b shows the system being worn after attachment of the electrodes.

The completed wearable EEG and ECG system weighs 42.5 g and its overall size is 158 × 142 × 90 mm^3^. The filter’s bandwidth was set as 0.5–50 Hz to measure each band (delta, theta, alpha, beta) of the EEG and to measure the R-peak of the ECG. Considering the filter bandwidth to avoid the aliasing effect, the sampling frequency was set to 250 Hz, which is also sufficient for the HRV analysis [43]. The ADC gain of the system was 12 V/V, and its AFE gain was 495 V/V for a total gain of 5940 V/V. Its input-referred dynamic range was ±407.4 μV, and it had a 24-bit ADC resolution. The noise level of the system was measured by shorting all electrodes, giving a noise level of <0.12 μVrms on three channels. The power consumption of the active mode that transmits data via Bluetooth is 9.6 mA, and so ~18 h of continuous use is possible using a 170 mAh Li-polymer battery.

### 3.2. EEG Alpha Wave and ECG Comparison Results

Figure 8a shows the raw waveforms of the left and right EEGs measured for 80 s in one subject. Every 20 s, the participant opened and closed his eyes, and the EEG amplitude shows an increase in the segments where the eyes were closed. Alpha waves occur dominantly when the majority of people close their eyes and meditate, and so the power spectrum was analyzed to confirm that the increased EEG amplitude was attributed to the alpha waves. Figure 8b shows the power spectrum results. As shown, the alpha wave frequency component of 8–13 Hz clearly increased when the eyes were closed. The EEGs of seven participants were analyzed, and the alpha waves were found to increase when the participants’ eyes were closed. These results confirm that the constructed wearable EEG and ECG system can measure two-channel EEGs.

The ECGs were measured simultaneously in the head and in both arms, which is the standard measurement method, and the results were compared. Figure 9a shows the raw waveforms of the two measured ECGs. The top waveform is the ECG measured in the head, which has an amplitude of ~25 μV, while the bottom waveform is the ECG lead I measured in both arms, which has an amplitude of ~0.5 mV. Therefore, to compare the morphologies of the two ECGs, the ECG measured in the head was amplified 20 times and overlaid as shown in Figure 9b. In particular, the waveforms were found to be similar in terms of the R-peak timing, from which the HRV could be extracted and used in the stress assessment. The R-peaks of each ECG were detected using the Pan and Tompkins algorithm to accurately compare the HRVs. The detected R-peaks are displayed over the ECG waveforms and compared, as shown in Figure 9c; the HRVs calculated from the R-peaks are compared in Figure 9d. Thirty R-peaks were measured for the seven participants, and the accuracy of the calculated HRVs was compared. All seven participants gave an accuracy of >99.5%, and so it was concluded that the HRV extracted from the ECG measured by the described wearable EEG and ECG system is sufficiently accurate. Additionally, we calculated six stress features, each from two different HRVs measured from the head, in addition to standard ECGs, and we performed an independent *t*-test. No significant difference was found in any of the features.

### 3.3. Stress Assessment

Table 2 shows the means and distributions of the EEG and HRV features of each section (S1, R1, S2, R2), as well as the results of an analysis of the statistically significant differences, which were carried out using ANOVA. Figure 10 shows the means and standard deviations of the features as a bar graph. As indicated, the APA was significantly smaller at S1 than at R1 and R2 (p < 0.001 and p = 0.005, respectively), and it was also smaller at S2 than at R1 and R2 (p < 0.001 and p = 0.002, respectively). This indicates that the right alpha power was reduced to a greater extent than the left alpha power in a stress situation, which is consistent with the physiological assumptions (i.e., enhanced activation occurred in the right hemisphere, which shows negative emotions). In addition, the LF/HF increased significantly more at S1 than at R1 and R2 (p = 0.045 and p = 0.02, respectively), and it also increased significantly more at S2 than at R1 and R2 (p = 0.001 and p < 0.001, respectively). Furthermore, although no significant difference was shown, the LF exhibited an increasing trend in the stress test, while the HF showed a decreasing trend. These results are in agreement with previous studies [2,13,17,44,45] demonstrating that in stress situations, the HF decreases while the LF and LF/HF increase. With the exception of the APA and LF/HF, none of the features showed significant differences between the stress (S1, S2) situations and the rest (S2, S3) situations. Figure 11 shows a box plot of the APA and LF/HF indicating the significant differences.

A linear SVM was then selected to develop a stress classifier model. Three models were developed, namely a model that uses only EEG features (EEG model), a model that uses only HRV features (HRV model), and a model that uses all features (EEG+HRV model). The features used in the models were selected in the order of the difference between the stress and rest periods. In the EEG model, four features were used: APA, nLAP, nRAP, and BPA, while the HRV model used LF/HF, nHF-HRV, nLF-HRV, and mRR, and the EEG+HRV model used nLAP, APA, nHF-HRV, and LF/HF. A five-fold cross-validation method was employed for each model. All samples were randomly divided into five groups. Four groups were used to build a model, and the remaining group was used as a validation group. This process was repeated five times by using a different validation group each time. The sensitivity, specificity, accuracy, and AUC were calculated to compare each model (Table 3). The EEG+HRV model was found to exhibit a superior performance to the EEG and HRV models (sensitivity: 90%, specificity: 85%, accuracy: 87.5%, AUC: 0.9564). It also demonstrated a superior performance when compared with previous stress classification studies [45,46]. It therefore appears that the simultaneous measurement of the EEG and HRV is essential for accurate stress assessment. Figure 12 shows the averaged receiver operating characteristics (ROC) curves of the five-fold cross-validation for the SVM classifiers of the EEG, HRV, and EEG+HRV models.

## 4. Conclusions

We herein reported our pioneering study for the development of a wearable system that can simultaneously measure two-channel electroencephalograms (EEG) and one-channel electrocardiograms (ECG). Importantly, the developed system is not only small and lightweight, it is also comfortable to wear and exhibits an excellent noise performance. The developed system was therefore verified by measurement of the EEG and ECG. More specifically, the left and right hemisphere EEGs were verified by alpha wave tests, and the ECG measured in the head was verified through comparison with an ECG lead I measured at the standard placement. The EEG and heart rate variability (HRV) features for stress assessments were extracted and a classification model was developed. It was found that the model employing both EEG and HRV features exhibited a superior performance, thereby confirming that the simultaneous measurement of the EEG and HRV improves the accuracy of stress assessments. We therefore expect that the developed wearable EEG/ECG system will contribute toward the sustained management of stress in daily life, which is important in modern society. In future studies, stress tests should be performed on a larger number of participants, including females. Furthermore, long-term monitoring during daily life is necessary to assess chronic stress, which may need a different classification algorithm with different feature sets. The wet electrodes should be replaced with dry electrodes to improve the convenience of wearing the device. 

## Figures and Tables

**Figure 1 sensors-19-01991-f001:**
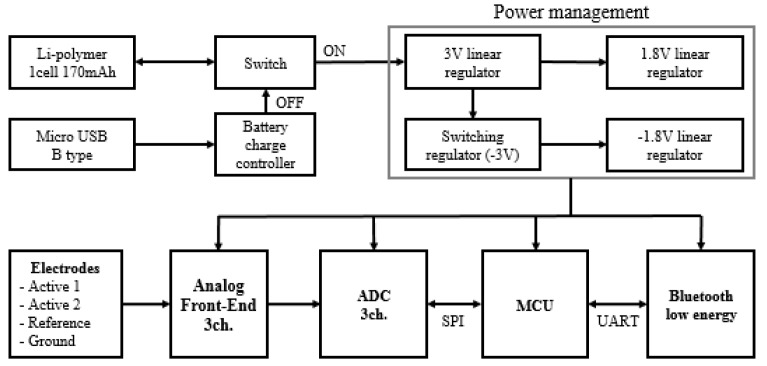
Circuit diagram of the wearable electroencephalogram (EEG) and electrocardiogram (ECG) recording system.

**Figure 2 sensors-19-01991-f002:**
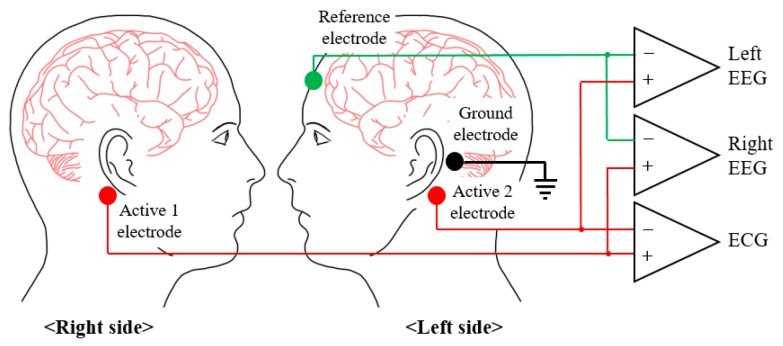
Placements and differential inputs of the electrodes for recording of the two EEG signals and one ECG signal (Active 1 – Active 2 = ECG, Active 1 − Reference = Right EEG, Active 2 − Reference = Left EEG).

**Figure 3 sensors-19-01991-f003:**
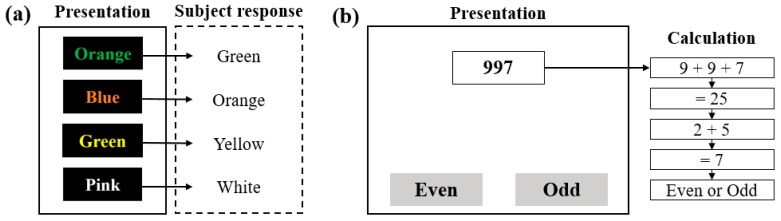
The methods for inducing stress: (**a**) the Stroop color word test, and (**b**) an illustration of the mental arithmetic test.

**Figure 4 sensors-19-01991-f004:**
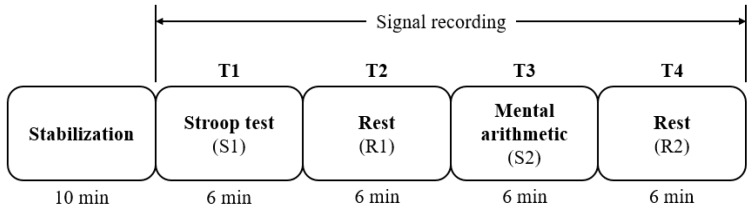
The stress test protocol employed herein.

**Figure 5 sensors-19-01991-f005:**
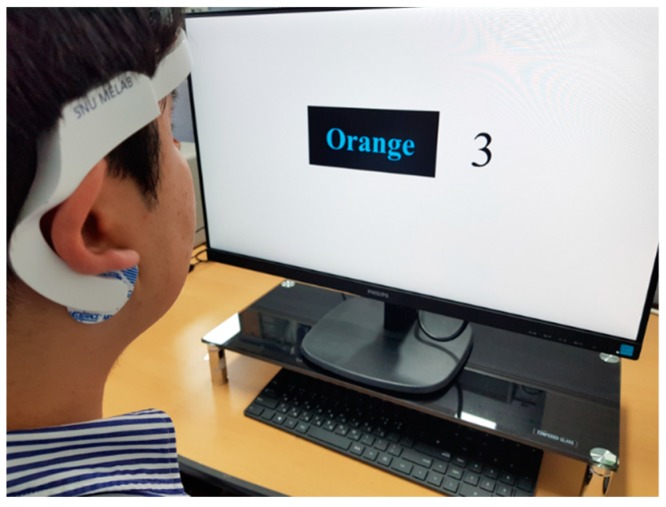
Photographic image of the experimental setup employed for the stress test.

**Figure 6 sensors-19-01991-f006:**
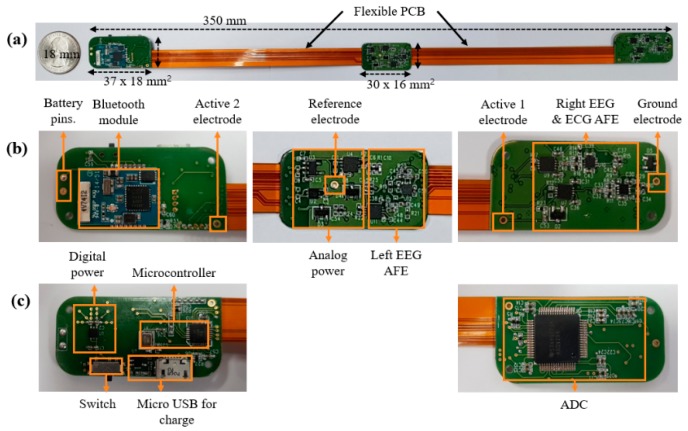
Photographic image of the printed circuit board of the wearable EEG and ECG recording system. (**a**) View of all printed circuit boards (PCBs), in which the three rigid PCBs are connected by flexible PCBs. (**b**) Enlarged view of the top side of the PCBs and their component layouts. (**c**) Enlarged view of the bottom side of the PCBs and their component layouts.

**Figure 7 sensors-19-01991-f007:**
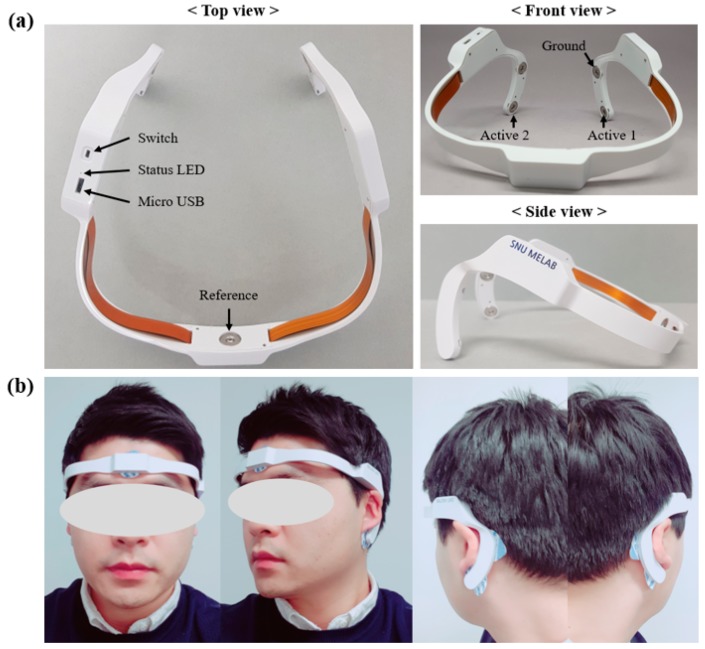
(**a**) The proposed wearable EEG and ECG system. (**b**) Photographic images of a subject wearing the system.

**Figure 8 sensors-19-01991-f008:**
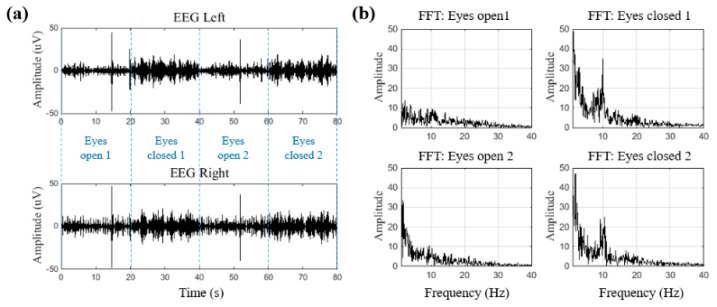
The alpha wave test: (**a**) EEG waveform of the eyes-open and eyes-closed sessions, and (**b**) power spectra of the eyes-open and eyes-closed sessions.

**Figure 9 sensors-19-01991-f009:**
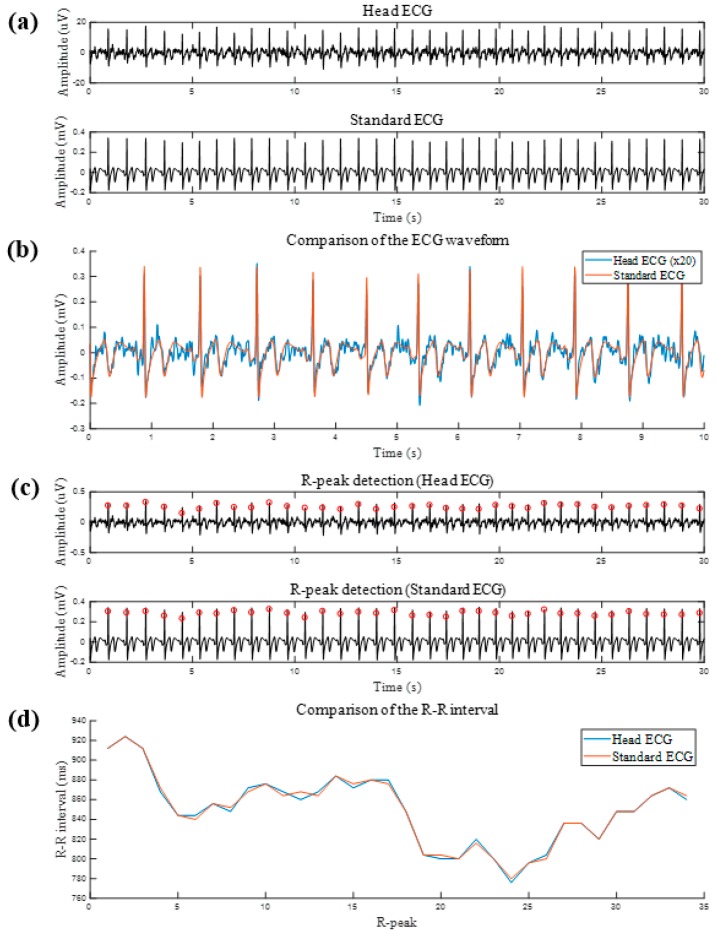
Comparison of the head ECG and the standard ECG. (**a**) Raw waveform of the head ECG and standard ECG. (**b**) Overlap of the head ECG (amplified 20 times) and the standard ECG. (**c**) R-peak detection of the head ECG and the standard ECG. (**d**) Comparison of the R–R interval from the head ECG and standard ECG.

**Figure 10 sensors-19-01991-f010:**
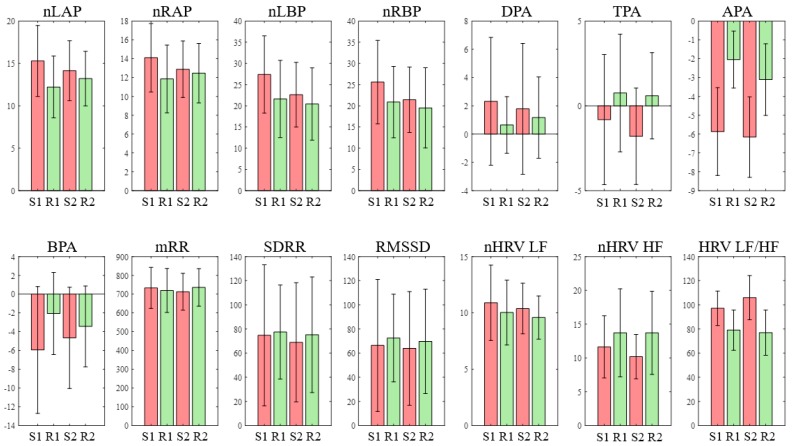
Mean values (standard error of mean, SEM) of fourteen features (S1 = Stroop test, R1 = rest1, S2 = mental arithmetic test, R2 = rest2).

**Figure 11 sensors-19-01991-f011:**
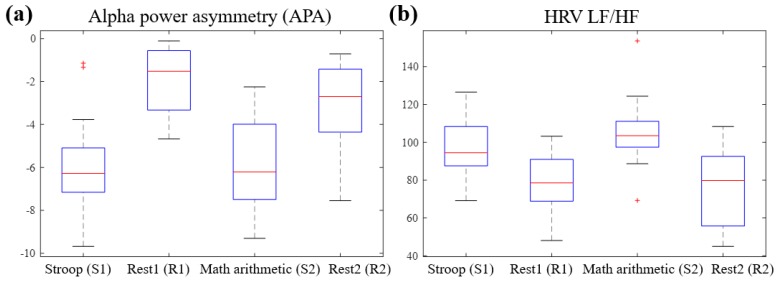
Box blots of features with significant differences between the stress and rest periods. (**a**) Alpha power asymmetry. (**b**) Heart rate variability (HRV) low-frequency power/high-frequency power (LF/HF). The boxes represent the 25th and 75th percentiles, the lines within the boxes represent the means, and the lines outside the boxes represent the most extreme data points not considered outliers. The plus symbol indicates outliers.

**Figure 12 sensors-19-01991-f012:**
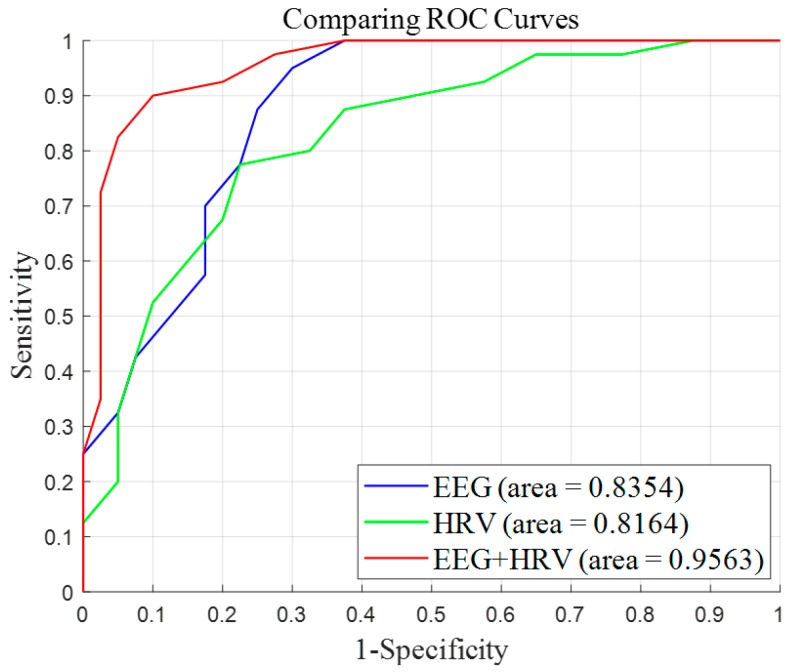
Average ROC curves of the five-fold cross-validation for the support vector machine (SVM) classifiers using the EEG, HRV, and EEG and HRV features. The blue line represents the EEG model (features: nLAP, nRAP, APA, BPA), the green line represents the HRV model (features: mRR, LF, HF, LF/HF), and the red line represents the EEG and HRV model (features: nLAP, APA, HF, LF/HF).

**Table 1 sensors-19-01991-t001:** Description of the electroencephalogram (EEG) and heart rate variability (HRV) features used in the stress assessments.

Feature	Description	Unit	Equation
nLAP	Normalized left hemisphere alpha band power	%	$\frac{\mathrm{Summation}\;\mathrm{of}\;\mathrm{power}\;\mathrm{from}\;8\;\mathrm{to}\;13\;\mathrm{Hz}}{Total\;power}\;\times 100$
nRAP	Normalized right hemisphere alpha band power	%	
nLBP	Normalized left hemisphere beta band power	%	$\frac{\mathrm{Summation}\;\mathrm{of}\;\mathrm{power}\;\mathrm{from}\;13\;\mathrm{to}\;30\;\mathrm{Hz}}{Total\;power}\;\times \;100$
nRBP	Normalized right hemisphere beta band power	%	
DPA	Delta band power asymmetry	-	$\frac{EEG\;band\;power_{R}-EEG\;band\;power_{L}}{EEG\;band\;power_{R}+EEG\;band\;power_{L}}$
TPA	Theta band power asymmetry	-	
APA	Alpha band power asymmetry	-	
BPA	Beta band power asymmetry	-	
mRR	Mean of R–R interval	msec	$\frac{\sum_{i=1}^{N}\left(RR_{i}\right)}{N}$
SDRR	Standard deviation of R–R interval	msec	$\sqrt{\left(\frac{\sum_{i=1}^{N}{\left(RR_{i}-mRR\right)}^{2}}{N-1}\right)}$
RMSSD	Root mean square difference of successive R–R interval	msec	$\sqrt{\frac{\sum_{i=1}^{N-1}{\left(RR_{i+1}-RR_{i}\right)}^{2}}{N-1}}$
nLF-HRV	Normalized low frequency power of HRV	%	$\frac{\mathrm{Summation}\;\mathrm{of}\;\mathrm{power}\;\mathrm{from}\;0.04\;\mathrm{to}\;0.15\;\mathrm{Hz}}{Total\;power}\;\times 100$
nHF-HRV	Normalized high frequency power of HRV	%	$\frac{\mathrm{Summation}\;\mathrm{of}\;\mathrm{power}\;\mathrm{from}\;0.15\;\mathrm{to}\;0.4\;\mathrm{Hz}}{Total\;power}\;\times 100$
LF/HF	The ratio between nLF-HRV and nHF-HRV	-	$\frac{nLF-HRV}{nHF-HRV}$

**Table 2 sensors-19-01991-t002:** Effects of the Stroop test and the mental arithmetic test on the EEG and HRV features.

Feature	Stroop Test (S1)	Rest (R1)	Arithmetic Test (S2)	Rest (R2)
nLAP	15.29 (±4.19)	12.22 (±3.64)	14.14 (±3.53)	13.21 (±3.22)
nRAP	14.11 (±3.63)	11.85 (±3.60)	12.88 (±3.00)	12.47 (±3.16)
nLBP	27.37 (±9.11)	21.62 (±9.09)	22.61 (±7.63)	20.42 (±8.53)
nRBP	25.60 (±9.83)	20.88 (±8.42)	21.43 (±7.71)	19.50 (±9.47)
DPA	2.32 (±4.52)	0.65 (±2.01)	1.78 (±4.63)	1.16 (±2.88)
TPA	−0.81 (±3.84)	0.76 (±3.47)	−1.80 (±2.85)	0.59 (±2.55)
APA	−5.87 (±2.33) ^1,2^	−2.05 (±1.51)	−6.16 (±2.14) ^1,2^	−3.11 (±1.90)
BPA	−5.95 (±6.76)	−2.07 (±4.37)	−4.66 (±5.40)	−3.44 (±4.31)
mRR	733.16 (±109.41)	719.31 (±116.85)	712.37 (±97.87)	735.24 (±100.15)
SDRR	74.74 (±58.37)	77.48 (±38.97)	68.93 (±49.41)	75.20 (±47.89)
RMSSD	66.36 (±54.69)	72.48 (±36.27)	63.94 (±47.09)	69.77 (±43.25)
nLF-HRV	10.89 (±3.35)	10.03 (±2.88)	10.38 (±2.25)	9.59 (±1.92)
nHF-HRV	11.63 (±4.59)	13.71 (±6.51)	10.20 (±3.29)	13.72 (±6.16)
LF/HF	97.15 (±14.26) ^1,2^	79.04 (±16.68)	105.93 (±18.34) ^1,2^	76.95 (±18.72)

Data are expressed as a mean ± standard deviation. ^1^ Significant differences between the stress test and rest1 (R1 versus S1 and R1 versus S2) (*p* < 0.05). ^2^ Significant difference between the stress test and rest2 (R2 versus S1 and R2 versus S2) (*p* < 0.05).

**Table 3 sensors-19-01991-t003:** Five-fold cross-validation results for the support vector machine (SVM) classifiers using EEG features (EEG model), HRV features (HRV model), and EEG and HRV features (EEG+HRV model)

Model	Sensitivity (%)	Specificity (%)	Accuracy (%)	AUC
EEG	84.6	72.0	77.9	0.8354
HRV	76.9	73.2	75.0	0.8164
EEG+HRV	90.0	85.0	87.5	0.9563

AUC: Area under the ROC curve.

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
