# Peer review of "A Novel Wearable EEG and ECG Recording System for Stress Assessment"

_sensors, 2019, doi:10.3390/s19091991_

Reviewer 1 Report

The article propose a novel sensor to measure both ECG and EEG for stress assessment. The work is well presented with good results supporting the use of this sensor for stress assessment. Still, some points still need to be adressed:

1) What is the sampling frequency for the ECG? 250Hz? While it should be enough for HRV analysis (you should support this decision with a reference, the European cardiology task force should work), most researchers prefer to use 500 or even 100 Hz.

2) From the RR-intervals, a resampling is needed to have a evenly sampled signal before doing a FFT. Which sampling frequency did you use for HRV?

3) You obtain a high accuracy when comparing the new ECG with the reference one. Did you also compute the HRV parameters from the reference ECG? There may be some significant differences in the parameters which were not present in the ECG signal from your sensor.

4) It is well known that respiration affects HRV. Indeed, it is difficult to interpret HRV when the person is speaking, since the respiration is not uniform. Did the subjects speak during the stroop test? If so, it should be noted in the discussion.   

Author Response

1) What is the sampling frequency for the ECG? 250Hz? While it should be enough for HRV analysis (you should support this decision with a reference, the European cardiology task force should work), most researchers prefer to use 500 or even 100 Hz.

Thank you for this point. We have added the text and relevant reference.

Page 7, Line 228

Considering the filter bandwidth to avoid the aliasing effect, the sampling frequency was set to 250 Hz, which is also sufficient for the HRV analysis [44].

44.    Electrophysiology Task Force of the European Society of Cardiology the North American Society of, P. Heart Rate Variability. Circulation 1996, 93, 1043-1065, doi:10.1161/01.CIR.93.5.1043.

2) From the RR-intervals, a resampling is needed to have a evenly sampled signal before doing a FFT. Which sampling frequency did you use for HRV?

As you commented, RR-intervals were resampled by 2Hz. We have added the text accordingly.

Page 5, Line 172

The HRV was lineally interpolated and resampled by 2 Hz to obtain uniformly spaced intervals, following which a fast Fourier transform (FFT) was performed to calculate the power spectrum for extracting the features in the frequency domain.

3) You obtain a high accuracy when comparing the new ECG with the reference one. Did you also compute the HRV parameters from the reference ECG? There may be some significant differences in the parameters which were not present in the ECG signal from your sensor.

We appreciate this comment. We have calculated six stress features each from  two different HRVs measured from the head as well as standard ECGs and performed an independent t-test. As shown in table below, no significant difference was found in all features.

mRR

SDRR

RMSSD

nLF-HRV

nHF-HRV

LF/HF

P-value

0.999

0.948

0.751

1.000

1.000

0.999

We have added the text accordingly.

Page 8, Line 264

Additionally, we calculated six stress features, each from two different HRVs measured from the head, in addition to standard ECGs, and we performed an independent t-test. No significant difference was found in any of the features.

4) It is well known that respiration affects HRV. Indeed, it is difficult to interpret HRV when the person is speaking, since the respiration is not uniform. Did the subjects speak during the stroop test? If so, it should be noted in the discussion.

As you commented, the HRV could be affected by the respiration. Therefore, in the stroop test, we asked the subject to think in the head without telling the color. Then, at the end of each stroop stimulus, we displayed the correct answer on the screen.

We have added the text accordingly.

Page 4, Line 132

The Stroop color word test is a reliable and valid method for inducing medium levels of stress [34,35]. In this test, the names of colors are shown in colors that are different from the colors corresponding to their names, and the subject must pick the colors that the names appear in (see Figure 3a). A total of 56 test stimuli were used, and they were each displayed randomly for 1 s. Because speaking can affect the HRV, the subjects were asked to think about the correct color, and the answer was displayed on the screen at the end of each stimulus.

Reviewer 2 Report

attached

Author Response

1. Introduction explained the background information about stress measurement using EEG and HRV. However, it lacks justification about why we use EEG and HRV together to measure stress. Also, the manuscript describes how to measure emotions using EEG, but there is no direct link between emotion and stress. The authors need to explain more clearly the scientific backgrounds.

We are sorry for the lacks of justification and reasoning.

Previous studies have revealed that each HRV and EEG is related to stress. Based on these finding, we have hypothesized that we can increase the accuracy of detecting stress by using both HRV and EEG like a previous study [31]. One of important goals in this study is to verify the hypothesis.

We have modified and added the text accordingly.

Page 2, Line 52.

Moreover, the simultaneous analysis of both the HRV and EEG is likely to provide a more precise assessment of stress [31] compared to the analysis of HRV alone, and one of the main objectives of this study is to verify this hypothesis.

We lacked a description of the relationship between emotion and stress. According to hemispheric specialization [23, 24], right prefrontal is more involved in negative emotion processing. Stress usually causes negative moods such as depression, anger and anxiety, resulting in increased right prefrontal activity [26]. Thus, several studies have assessed the stress through asymmetric analysis of frequency band powers in the EEG [20, 26-30].

We have modified and added the text accordingly.

Page 2, Line 49.

Stress usually causes negative moods, such as depression, anger, and anxiety, resulting in increased right-prefrontal activity [26]. Thus, the asymmetric analysis of the frequency-band powers in the EEG measured at the prefrontal cortex has been generally applied in previous stress studies [20,26-30].

2. Line 77-84: There is no information about the microcontroller used in this study. There is no information about the filters’ cutoff frequencies.

Thank you for this comment. We have added the text.

Page 2, Line 88: and an ATmega168 microcontroller (Microchips Technology Inc., USA) was used for the system control.

Page 2, Line 84: The cut-off frequencies of the HPF and LPF were set to 0.5 Hz and 50 Hz, respectively.

3. It is better to show circuit diagram in Figure 1, and the photo images in Figure 1 are better to show in result section.

As you recommended, we added a circuit diagram (Figure 1) to the method and moved the existing PCB image (Figure 6) to result.

We have added the text and figure accordingly.

Page 2, Line 89.

The overall circuit diagram of the EEG and ECG recording system is shown in Figure 1.

Figure 1. Circuit diagram of the wearable EEG and ECG recording system.

4. Line 130: Why the study used number of peaks (30 R-peaks) rather than time (e.g., 30 sec)?

Each subjects has different heart rate and therefore has different number of R-peaks within a fixed time range. To compare all subjects with same number of RR-intervals, we have selected the number of R-peaks instead of the fixed time range.

5. Line 132: Why there is no woman among the participants?

All subjects volunteered for this study were male coincidentally. We have included this issue as a future study. Thank you for this comment.

Page 11, Line 339.

In future studies, stress tests should be performed on a larger number of participants, including females.

6. Line 141- 142: Needs a reference for “The mental arithmetic test is a standard moderate intensity stressor used in physiology to detect changes in the ANS function”

As you recommended, we added a reference [38] about the mental arithmetic test.

Page 4, Line 139.

38.  Schneider, G.M.; Jacobs, D.W.; Gevirtz, R.N.; O'Connor, D.T. Cardiovascular hemodynamic response to repeated mental stress in normotensive subjects at genetic risk of hypertension: evidence of enhanced reactivity, blunted adaptation, and delayed recovery. Journal of Human Hypertension 2003, 17, 829, doi:10.1038/sj.jhh.1001624.

7. Figure 8(d) need an index.

We are sorry for the missing. Previous figure 8 is changed to the figure 9. We added an index to Figure 9(d)

Page 8, Line 266.

8. What are the values for Gamma, degree and C (polynomial kernel is chosen with penalty parameter) in SVM algorithm?

We used the linear SVM algorithm as written in the line 300 and C parameter was set to one.

9. Line 288: How many folds is used for test and how many is used as train?

We have added the information in our manuscript.

Page 10, Line 307.

A five-fold cross-validation method was employed for each model. All samples were randomly divided into five groups. Four groups were used to build a model, and the remaining group was used as a validation group. This process was repeated five times by using a different validation group each time.

10. Discussion point: The proposed device was evaluated by inducing acute stress in this study. However, the wearable device will be more useful to monitor chronic stress rather than acute stress. We may need different feature sets and algorithms to assess chronic stress. Any thought?

We appreciate this valuable comment. We totally agree with your comments. The developed system has particular advantages in long-term stress monitoring in daily life. Thus, this point is added as a future study. Thank you again for this point.

We have modified and added the text accordingly.

Page 11, Line 340.

Furthermore, long-term monitoring during daily life is necessary to assess chronic stress, which may need a different classification algorithm with different feature sets.

Round  2

Reviewer 2 Report

The revised manuscript addressed all of my concerns and questions. I do not have further comments.